# Qinghai Province (Tibetan Plateau): Quantifying the Influence of Climate Change and Human Activities on Vegetation Net Primary Productivity and Livestock Carrying Capacity Growth Potential

**DOI:** 10.3390/biology14050494

**Published:** 2025-05-01

**Authors:** Qian Wei, Bingrong Zhou, Wenying Wang

**Affiliations:** 1College of Geographic Sciences, Qinghai Normal University, Xining 810016, China; weiqian1024@163.com; 2Qinghai Provincial Meteorological Bureau, Xining 810008, China; zbr0515@foxmail.com; 3Plateau Science and Sustainable Development Research Institute, Xining 810016, China; 4Key Laboratory of Biodiversity Formation Mechanism and Comprehensive Utilization in the Qinghai Tibet Plateau, Xining 810016, China

**Keywords:** CASA model, quantitative analysis, driving mechanisms, ecological management

## Abstract

Understanding how climate change and human actions affect plant growth in Qinghai Province (located on the Tibetan Plateau) is crucial for protecting its fragile ecosystems and supporting sustainable livestock farming. However, little research has been conducted in this region. Our study investigates how shifts in temperature, rainfall, and human activities have influenced plant productivity over nearly four decades (1982–2018). We also assess how much more livestock the land could sustainably support. By comparing two scientific models, we found that climate factors boosted plant growth in 87% of Qinghai, particularly in its eastern, central, and southern regions. Human activities caused plant decline in 11% of the area, mainly in the northwest. Over time, the land’s potential to support livestock improved steadily, with eastern Qinghai showing the highest gains. Our findings provide a science-based roadmap for balancing ecological health and economic needs in Qinghai, benefiting local communities and policymakers working toward sustainable land use.

## 1. Introduction

Vegetation net primary productivity (NPP) quantifies the organic matter accumulated by vegetation per unit area. It denotes the total primary productivity gained through photosynthesis, subtracting the portion consumed by plant respiration [1,2]. NPP serves as the fundamental material and energy basis for the Earth’s life system, underpinning the genesis and sustenance of various ecosystem services [3]. Formed through intricate plant–environment interactions, NPP not only mirrors the production capacity and ecological quality of green plant communities but also holds pivotal roles in ecological dynamics and carbon cycling. It stands as a crucial scientific indicator, evaluating the collective impact of land ecosystems on atmospheric carbon exchange, climate shifts, and human interventions on vegetation [4].

Studies reveal substantial variations in vegetation net primary productivity (NPP) dynamics and influencing factors across diverse regions and spatiotemporal scales [5,6]. Globally, from 2000 to 2010, 49.25% of grassland ecosystem NPP witnessed a decline. Climate-induced soil degradation accounted for 45.51% of the decrease, and human activities contributed to 32.53%. Additionally, 39.40% of increased grassland NPP was attributed to human disturbances, while 30.6% of recovery resulted from climate change, with Asia experiencing the most significant NPP degradation and recovery [7]. In China, influenced by climate warming, land vegetation NPP displays a positive correlation with precipitation in most areas, showcasing a growth trend due to climate change. Some regions exhibit a positive correlation with average temperature, albeit weaker than that with precipitation [8,9,10,11,12]. Research on the Qinghai–Tibet Plateau by Chen et al. [13] unveils spatiotemporal variations in the relative impacts of climate change and human activities on vegetation NPP, forming a “four lines-five zones” pattern. Liu et al. [14] examine Qinghai Province, attributing spatial–temporal changes in vegetation NPP to climate factors and altitude. Yang et al. [15] quantify climate change and human activities’ effects on Qilian Mountains National Park’s vegetation NPP, with climate change explaining 92% of recovery and human activities contributing to 71% of degradation. While prior studies employ factors like temperature and precipitation, using correlation analyses to describe NPP spatial differentiation, there remains a gap in quantifying climate change and human activity impacts on ecosystems [16]. Qinghai Province, in particular, lacks sufficient research in this regard.

Quantifying variations in vegetation NPP and discerning driving factors in the study area are pivotal for understanding the mechanisms steering vegetation changes, unraveling the impacts of climate change and human activities on NPP, and effectively managing the ecological environment [15]. Recent studies investigating the causes of vegetation NPP changes, attributed to climate change and human activities, have employed various methods, including the coefficient of variation, geographical detector, regression analysis, and the disparity between actual NPP and potential NPP. The latter method, assessing the relative effects of human activities, calculates the difference between actual and potential productivity. Potential productivity represents the maximum NPP achievable under natural conditions, predominantly influenced by temperature and precipitation. In contrast, actual productivity encapsulates the combined impacts of climate change and human activities on vegetation NPP. This method, known for its clear biological significance and simplicity, has emerged as a primary approach for quantitatively studying the driving forces behind vegetation NPP [16,17,18,19,20,21,22].

In this study, we juxtapose climate-driven potential NPP changes, derived from the Zhou Guangsheng model, with actual NPP changes computed using the Carnegie–Ames–Stanford Approach (CASA, https://www.nasa.gov/casa-homepage accessed on 21 May 2022) model in Qinghai Province. The objective is to scrutinize vegetation recovery and degradation dynamics, discern the relative influences of climate change and human activities on vegetation alterations, quantitatively analyze the predominant factors steering NPP changes across different Qinghai regions, precisely evaluate the impacts of climate change and human activities on diverse ecological environments, and identify the potential for carrying capacity growth. This comprehensive analysis aims to establish a theoretical foundation for understanding the intricate connections between regional ecosystems and environmental factors. Moreover, it seeks to contribute insights for enhancing the ecological environment and fostering the judicious development and utilization of natural resources in Qinghai Province.

## 2. Research Area and Methods

### 2.1. Overview of the Study Area

Qinghai Province, situated in the eastern part of the Qinghai–Tibet Plateau, plays a vital role as the headwater region for three major rivers: the Yellow River, Yangtze River, and Lancang River. Spanning from longitude 89°35′ to 103°04′ east and latitude 31°40′ to 39°19′ north, it encompasses a vast area of 72.23 × 10^4^ km^2^ [23]. The topography follows a pattern of high terrain in the south and north, a central low region, and elevated areas in the west contrasting with lower terrain in the east, featuring elevations exceeding 3000 m. Positioned inland with intricate topography, the region experiences a plateau continental climate, characterized by low and concentrated precipitation due to minimal monsoonal influence. With a total natural grassland area of 4191 × 10^4^ hm^2^, of which 3864.58 × 10^4^ hm^2^ is usable, grassland covers 47% of the land area, forming a crucial foundation for livestock production. This establishes Qinghai as one of China’s five major pastoral regions [24,25]. The province is administratively divided into 44 county-level regions, encompassing 6 districts, 27 counties, 7 autonomous counties, and 4 county-level cities. Figure 1 illustrates the distribution of counties in the research area, and the complete names of the county abbreviations can be referenced in Appendix A.

### 2.2. Data Sources and Preprocessing

(1)Remote Sensing Data and Preprocessing:

The remote sensing vegetation index data utilized in this study were sourced from the China Scientific Data Platform (http://data.tpdc.ac.cn/zh-hans/ accessed on 18 May 2022). Specifically, this study employed the 1982–2020 monthly NDVI dataset for China, featuring a spatial resolution of 5 km. To focus on Qinghai Province, a monthly NDVI series dataset spanning 1982–2020 was extracted, employing a vector data mask outlining the Qinghai boundary. This rigorous preprocessing ensures the dataset’s alignment with this study’s geographical scope and facilitates accurate analysis of vegetation dynamics within Qinghai.

(2)Meteorological Data and Preprocessing:

The meteorological data employed in this study were furnished by the Qinghai Institute of Meteorological Sciences. This dataset encompasses monthly average temperature, monthly precipitation rate, and monthly downward shortwave radiation grid data spanning 1982 to 2018 in Qinghai Province. To ensure compatibility with the NDVI dataset, these three meteorological variables underwent meticulous preprocessing steps. The data were subjected to masking, unit conversion, projection alignment, and resampling to achieve a uniform spatial resolution and projection, establishing a cohesive Qinghai Province meteorological data grid dataset for accurate and synchronized analysis.

(3)Vegetation Type Data:

The vegetation type data utilized in this study originated from the GlobeLand30-2010 dataset, accessed through the National Geospatial Information Resources Catalog Service System. This dataset, boasting a spatial resolution of 30 m, encompasses ten primary land cover types: cropland, forest, grassland, shrubland, wetland, water bodies, tundra, artificial surfaces, bare land, glaciers, and permanent snow. To align with the NDVI data, the Qinghai Province vegetation type data underwent a meticulous process. This involved mosaicking and resampling strips covering Qinghai Province and subsequent masking, ensuring uniform spatial resolution for comprehensive integration and analysis.

(4)Station Data:

Field measurements crucial to this study were gathered from monitoring stations overseen by the meteorological department of Qinghai Province. These reliable station data form a cornerstone in augmenting the accuracy and depth of our analysis, providing on-the-ground insights into meteorological parameters essential for a comprehensive understanding of the region’s dynamics.

### 2.3. Research Methods

(1)Vegetation NPP Calculation Method

The CASA model has advantages such as high accuracy and ease of operation compared to other models. Zhang Yonghong [26,27] conducted a qualitative study on NPP in Qinghai Province using the CASA model. In this study, we adopted the CASA model, refined by Zhu Wenquan et al., drawing insights from the research of Zhang Yonghong for NPP estimation. The CASA model relies on two key variables to quantify NPP: absorbed photosynthetically active radiation (APAR) by vegetation and actual light use efficiency (ε). The calculation formula is expressed asNPP(x,t) = APAR(x,t) × ε(x,t)

Here, APAR(x,t) signifies the absorbed photosynthetically active radiation (gC·m^−2^·month^−1^) at location x in month t, while ε(x,t) denotes the actual light use efficiency (gC·MJ^−1^) at the same location and time. This formula forms the basis for our comprehensive assessment of vegetation NPP dynamics.

(i)Estimation of APAR

APAR is derived as the product of total solar radiation and the fraction of incident radiation absorbed by vegetation, defined by the following calculation formula:

APAR(x,t) = SOL(x,t) × FPAR(x,t) × 0.5

Here, SOL(x,t) symbolizes the total solar radiation (MJ·m^−2^·month^−1^) at location x in month t, and FPAR(x,t) denotes the fraction of incident radiation absorbed by vegetation. The constant 0.5 signifies the proportion of solar radiation utilized by vegetation relative to the total solar radiation, providing a comprehensive representation of absorbed photosynthetically active radiation in our analysis.

(ii)Estimation of FPAR

There exists a robust linear correlation within a specified range between FPAR and both the Normalized Difference Vegetation Index (NDVI) and the Simple Ratio (SR). The calculation formulas are as follows:$$FPAR(x,t)=\frac{NDVI(x,t)-{NDVI}_{i,min}}{{NDVI}_{i,max}-{NDVI}_{i,min}}\times {(FPAR}_{max}-{FPAR}_{min})+{FPAR}_{min}$$$$FPAR(x,t)=\frac{SR(x,t)-{SR}_{i,min}}{{SR}_{i,max}-{SR}_{i,min}}\times {(FPAR}_{max}-{FPAR}_{min})+{FPAR}_{min}$$
where FPAR_min_ and FPAR_max_ have values of 0.01 and 0.95, respectively. NDVI_i,min_ has a value of 0.035, while NDVI_i,max_ ranges from 0.634 to 0.7785. SR_i,min_ has a value of 1.0725, while SR_i,max_ ranges from 4.46 to 8.03 [26].To obtain SR(x,t):$$SR(x,t)=\frac{1-NDVI(x,t)}{1+NDVI(x,t)}$$

The estimation of FPAR is derived by averaging the estimation results from FPAR-NDVI and FPAR-SR. The calculation formula is as follows:$$FPAR(x,t)={(\alpha FPAR}_{NDVI}+(1-\alpha ){FPAR}_{SR})$$
where α represents the adjustment coefficient between the two methods, uniformly set to 0.5 in this study, reflecting an equal weighting of the two methods.

(iii)Estimation of light use efficiency (ε)

Light use efficiency (ε) is the ratio of all chemical potential energy contained in the biomass produced per unit area over a given time period to the photosynthetically active radiation (PAR) absorbed. Environmental factors like temperature, soil moisture conditions, and atmospheric vapor pressure deficit regulate vegetation NPP by influencing plant photosynthesis. In remote sensing models, these factors are adjusted by modifying the maximum light use efficiency. The calculation of light use efficiency is as follows [27]:$$\varepsilon (x,t)={(T}_{\varepsilon 1}(x,t)\times T_{\varepsilon 2}(x,t)\times W_{\varepsilon }(x,t)\times \varepsilon_{max})$$

In the equation, T_ε1_(x,t) and T_ε2_(x,t) represent the coefficients of low temperature and high temperature stress, respectively. W_ε_(x,t) represents the coefficient of water stress. ε_max_ is the maximum light use efficiency under ideal conditions. The specific calculation methods for these coefficients can be found in the research conducted by Zhang Yonghong [5].

(2)Calculation Method for Climate NPPP

Zhou Bingrong et al. extensively evaluated five climate models for estimating vegetation NPP in the Sanjiangyuan region. Their findings highlighted the superior accuracy and effectiveness of the model developed by Zhou Guangsheng [26]. Hence, in this study, we opted for the well-established natural vegetation net primary productivity (NPP_P_) model crafted by Zhou Guangsheng et al., as detailed in the research conducted by Zhang Xinshi et al. [26].$${NPP}_{P}=RDI\cdot \frac{r\cdot (1+RDI+{RDI}^{2})}{\left(1+RDI)(1+{RDI}^{2}\right)}\cdot Exp(-\sqrt{9.87+6.25RDI})$$$$RDI={\left(0.629+0.273PER-0.00313{PER}^{2}\right)}^{2}$$PER=PET/r=BT·58.93/rBT=∑T/12

In the equation, RDI represents the radiation dryness index, PER represents the potential evapotranspiration rate, PET represents the annual potential evapotranspiration (mm), r represents the annual rainfall (mm), BT represents the annual average biological temperature (°C), and T represents the monthly average temperature between >0 °C and <30 °C.

NPP_H_ represents the NPP loss or gain caused by human activities. The calculation formula is as follows:$${NPP}_{H}={NPP}_{A}-{NPP}_{P}$$

(3)Analysis of Interannual Variation Trend in Vegetation NPP

The method of least squares is utilized to calculate the slope of NPP changes at the pixel level, providing insights into the interannual variation trend. This analysis captures whether NPP shows an upward or downward trend over time, reflecting the vegetation’s recovery or degradation status. Additionally, the spatial distribution pattern of NPP may undergo changes. The formula for this analysis is as follows:$$S=\frac{n\cdot \sum_{i=1}^{n}(i\cdot V_{i})-\sum_{i=1}^{n}i\cdot \sum_{i=1}^{n}V_{i}}{n\cdot \sum_{i=1}^{n}i^{2}-{\left(\sum_{i=1}^{n}i\right)}^{2}}$$

The formulas provided are as follows:S: Linear trend value;V_i_: NPP_A_, NPP_P_, or NPP_H_;i: Year index; n = 37 (referring to the number of years);S > 0 indicates an increasing trend in NPP;S < 0 indicates a decreasing trend in NPP.

The absolute value of S reflects the rate of increase or decrease in NPP.

Significance testing of different NPP trends employs the correlation coefficient method. For a sample size of n = 37, r values falling within [0.418, 1] signify a highly significant positive correlation, while values within [0.325, 0.418) indicate a significant positive correlation. Correlation is considered insignificant when r values lie within (−0.325, 0.325), and values within (−0.325, −0.418] suggest a significant negative correlation. Highly significant negative correlation corresponds to r values within the range [−1, −0.418].

Based on the calculated linear trend values (S) and the results of the significance tests, NPP trends can be classified into six categories:Highly significant decrease (S < 0, *p* ≤ 0.01);Significant decrease (S < 0, 0.01 < *p* ≤ 0.05);Insignificant decrease (S < 0, *p* > 0.05);Insignificant increase (S > 0, *p* > 0.05);Significant increase (S > 0, 0.01 < *p* ≤ 0.05);Highly significant increase (S > 0, *p* ≤ 0.01).

Performed univariate linear regression analyses and significance tests to uncover the spatial distribution characteristics of NPP_A_, NPP_P_, and NPP_H_ trends over time. Calculated the percentage change in these variables to enhance understanding of their evolving patterns in Qinghai Province.

(4)Quantitative Evaluation of the Impact of Climate Change and Human Activities on Vegetation NPP

To distinguish the impacts of climate change and human activities on vegetation NPP recovery and degradation, linear trend values are calculated for NPP_A_, NPP_P_, and NPP_H_, represented as SA, SP, and SH, respectively. A positive SA suggests vegetation actual productivity in a recovering state, while a negative SA indicates vegetation actual productivity in a degrading state. The quantitative evaluation method encompasses five distinct scenarios, outlined in Table 1.

(5)The model-simulated net primary productivity (NPP) of vegetation requires validation using the site observation data provided by the meteorological department in Qinghai Province. Compare the NPP values calculated using the CASA model with the actual site values to verify if they match.(6)The CASA model simulates the net primary productivity (NPP) of vegetation, which needs conversion into edible hay yield for calculating the grass–livestock balance status. This is achieved using the following formula:

GY = NPP/[1 + (BNPP/ANPP)], NPP = ANPP + BNPP

Here, GY represents the grass yield (hay yield), BNPP represents belowground net primary productivity, and ANPP represents aboveground net primary productivity. NPP is the annual net primary productivity measured in gC/m^2^/a (grams of carbon per square meter per year). Based on the literature, the ratio of BNPP to ANPP in Qinghai region is approximately 4 [28]. The carbon content of aboveground vegetation is assumed to be 0.43 [29], which is used to convert NPP into dry matter weight. Therefore, the formula for calculating hay yield in Qinghai Province, based on biomass dry weight, isGY__2_ = NPP/5/0.43
where GY__2_ represents aboveground dry hay weight in gDM/m^2^/a (grams of dry matter per square meter per year).

The theoretical carrying capacity of grassland can be calculated using the following formula:Cl = Y × K × U/(R × T)
where Cl represents the theoretical carrying capacity of grassland, Y represents the available amount of grass per unit area measured in kg/hm^2^, K represents the proportion of edible forage in the grassland (assumed as 80%), U represents the utilization rate of grassland (assumed as 50%), R represents the daily forage intake of a standard sheep unit (calculated as 1.5 kg of dry weight per day) [29], and T represents the grazing days (assumed as 365 days).

The potential for increasing livestock carrying capacity can be determined by subtracting the actual theoretical carrying capacity from the carrying capacity under climatic potential conditions.

## 3. Results and Analysis

### 3.1. Model Verification

In this study, we validated the actual net primary productivity (NPP) values simulated by the CASA model against measured data from 120 grassland samples within and outside the fences of 20 ecological monitoring sites in Qinghai Province spanning 2010 to 2015. Figure 1 illustrates the comparison, revealing the optimal simulation outcomes when contrasting CASA model values with measurements inside the fences (Figure 2a). The linear regression slope stands at 0.5732, accompanied by an R-squared value of 0.471. The second-best results are observed when comparing CASA model values with measurements outside the fences (Figure 2b), featuring a linear regression slope of 0.866 and an R-squared value of 0.3633. These findings underscore the robust agreement between simulated NPP values from 2010 to 2015 and the measured values. The enhanced CASA model demonstrates high simulation accuracy and reliability in estimating the actual NPP of vegetation in Qinghai Province.

### 3.2. Spatial Distribution Characteristics of NPP_A_ at County Scale in Qinghai Province

The average vegetation net primary productivity (NPP_A_) in Qinghai Province from 1982 to 2018, influenced by latitude zonality, longitude zonality, vegetation, climate, and other factors, demonstrates a distinct spatial pattern. Notably, higher NPP_A_ values are observed in the eastern regions, contrasting with lower values in the west (Figure 3). Counties with an average vegetation NPPA exceeding 300 g C/m^2^/year during the same period encompass Henan, Zeku, Tongde, Tongren, Xunhua, Gande, Jiuzhi, Baima, Yushu, Nangqian, Gangcha, Haiyan, Qilian, Datong, Huangyuan, and Huangzhong. These counties, primarily situated in the eastern part of Qinghai Province, exhibit elevated NPP_A_ levels. Meanwhile, counties with NPP_A_ values ranging from 100 to 300 g C/m^2^/year are concentrated in the central part, while those with NPP_A_ values below 100 g C/m^2^/year are predominantly located in the western part of Qinghai Province. This spatial distribution underscores the nuanced interplay of geographical and environmental factors shaping vegetation productivity across the province.

### 3.3. Trends in NPP_A_, NPP_P_, and NPP_H_

Statistical analysis was employed to assess the average annual values of vegetation net primary productivity (NPP_A_), climate potential NPP_P_ (NPP_P_), and human-induced NPP (NPP_H_) in Qinghai Province for each year, and the results were subjected to linear regression against the year (Figure 4). Temporally, the average vegetation NPP_A_ exhibited an ascending trajectory from 1982 to 2018. The annual average NPP_A_ per unit area increased from 123.77 g C/m^2^/year in 1982 to 163.37 g C/m^2^/year in 2018, demonstrating a growth rate of 1.0476 g C/m^2^/a. Concurrently, the average NPP_P_ demonstrated an upward trend, with the annual average climate potential NPP_P_ per unit area rising from 279.02 g C/m^2^/year in 1982 to 329.43 g C/m^2^/year in 2018, at a growth rate of 2.8973 g C/m^2^/a. Conversely, the average NPP_H_ showcased a declining trend, decreasing from −113.25 g C/m^2^/year in 1982 to −116.05 g C/m^2^/year in 2018, indicating a rate of −1.8497 g C/m^2^/a.

From 1982 to 2018, the change in NPP_A_ in Qinghai Province (Figure 5) primarily witnessed an increase, encompassing 88.6% of the total land area. Within this, 63% exhibited a significant increase (*p* < 0.01), while 14% showed a notable increase (*p* < 0.05). Regions with no significant changes constituted 23% of the total area. The significantly increasing areas were concentrated in the eastern part, while those with notable increases were mainly distributed in the southern part. Areas depicting a decreasing trend in NPP_A_ covered 11% of the total land area, with 73.5% exhibiting a significant decrease (*p* < 0.01), primarily located in the northwestern part, including Golmud, Mangya, and Dulan counties.

From 1982 to 2018, the trend of NPP_P_ in Qinghai Province (Figure 6) predominantly manifested as an increase, covering 99% of the total land area. Within these regions, 67% exhibited a significant increase (*p* < 0.01), while 15% displayed a notable increase (*p* < 0.05). Areas with no significant changes constituted 18% of the total area. The significantly increasing regions were concentrated in the eastern and central parts of Qinghai Province. In contrast, regions depicting a decreasing trend in NPPP accounted for 1% of the total land area, primarily concentrated in Mangya county in the northwestern part of Qinghai Province, with 100% of this region showing no significant changes.

From 1982 to 2018, the trend of NPP_H_ in Qinghai Province (Figure 7) primarily exhibited a decrease, covering 90% of the total land area. Among these regions, 52% demonstrated a significant decrease (*p* < 0.01), while 18% displayed a notable decrease (*p* < 0.05). Areas with no significant changes constituted 30% of the total area. The significantly decreasing regions were mainly distributed in the central and eastern parts of Qinghai Province. In contrast, regions depicting an increasing trend in NPP_H_ covered 10% of the total land area, with 5% demonstrating a significant increase (*p* < 0.01) and 9% exhibiting a notable increase (*p* < 0.05). The significantly increasing regions were distributed in counties such as Tianjun, Gonghe, Datong, Jiuzhi, Banma, and Yushu, primarily in the northeastern and southern parts of Qinghai Province.

Figure 8 illustrates the distribution map of driving factors for vegetation NPP changes in Qinghai Province. This determination is made by comparing the absolute values of the slope (S) of NPP_P_ and NPP_H_ changes at each pixel, utilizing the discrimination method outlined in Table 1. The map categorizes into five types: no change, NPP increase due to climate factors, NPP increase due to human activities, NPP decrease due to climate factors, and NPP decrease due to human activities. This detailed classification provides insights into the nuanced influences shaping vegetation NPP dynamics across the province.

Figure 8 reveals that from 1982 to 2018, the restorative impact of climate change and human activities on vegetation in Qinghai Province surpassed the degradative effect. Regions witnessing increased vegetation NPP covered 87% of the total area, with 73% attributed to climate factors, spanning the entire province and concentrated in the east, center, and south. The remaining 27%, influenced by human activities, were scattered in counties like Tianjun, Ulan, Gonghe, Jiuzhi, and Banma, primarily in the northeastern and southern parts. Areas experiencing decreased vegetation NPP constituted 11% of the total area, with 1% attributed to climate factors in the northwest of Mangya County and 99% to human activities, mainly in Golmud, Mangya, and Dulan counties in the northwest. Unchanged vegetation NPP areas comprised 2% of the total area. Climate factors played a predominant role in both restoration and degradation, accounting for 64.6% of the total area, while human activities contributed to 34.3%.

### 3.4. Livestock Carrying Capacity Growth Potential

Figure 9 illustrates a temporal increase in climatic carrying capacity for livestock in Qinghai Province from 1982 to 2018. The average annual climatic carrying capacity per unit area rose from 0.81 SHU/ha in 1982 to 1.12 SHU/ha in 2018. Similarly, the theoretical carrying capacity demonstrated an upward trajectory during this period, with the average annual theoretical carrying capacity per unit area increasing from 0.42 SHU/ha in 1982 to 0.56 SHU/ha in 2018. Additionally, the potential for livestock carrying capacity growth in Qinghai Province exhibited an upward trend, with the average annual potential for livestock carrying capacity per unit area increasing from 0.38 SHU/ha in 1982 to 0.56 SHU/ha in 2018.

In 2018, as shown in Figure 10, on a spatial scale, regions in Qinghai Province with a positive potential for livestock carrying capacity growth covered 95% of the total area. Among these, areas with high livestock carrying capacity and growth potential, ranging from 1 to 2.1 SHU/ha, were distributed in counties such as Huzhu, Ledu, Hualong, Minhe, Ping’an, Huangzhong, Xunhua, Gonghe, Guide, Guinan, Tongde, Xinghai, Ulan, Dulan, and Maqin, encompassing 9% of the total area and primarily located in the eastern part of Qinghai Province. Conversely, regions with a negative potential for livestock carrying capacity growth constituted 5% of the total area, mainly concentrated in Mangya County in the northwestern part of Qinghai Province.

## 4. Discussion

### 4.1. Relationship Between NPP_A_ and Human Activities

Human activities in Qinghai Province exhibit a dual impact on net primary productivity (NPP), both positively promoting and negatively inhibiting it. Policies such as the Qinghai Grass–Livestock Balance Policy, grassland ecosystem protection subsidies, and the “Grassland dedicated person responsibility system” have significantly contributed to grassland ecosystem protection and utilization. These policies have increased vegetation coverage, directly accelerating the growth rate of vegetation NPP [30,31,32]. This study indicates that 27% of the increase in vegetation NPP is attributed to human activities. Areas where human activities drive the increase in vegetation NPP, like Gonghe county, are linked to major ecological projects and forage planting initiatives [33]. However, human activities also play a major role in the decrease of vegetation NPP in Qinghai. This study reveals that 11% of Qinghai Province experiences a reduction in vegetation NPP, with 99% of this area caused by human activities, concentrated in Golmud, Mangya, and Duolun counties in the northwest. This is associated with western development, urbanization, city expansion, and population increase since the Fifteenth Five-Year Plan period [34]. Climate warming contributes to population migration, impacting human activities and, subsequently, vegetation degradation [35]. The decrease in vegetation NPP_A_ in the northwest may also be related to vegetation belt degradation caused by climate warming. Implementing proactive ecological construction and protection measures is crucial for controlling and mitigating vegetation degradation in the future.

### 4.2. Relationship Between NPP_A_ and Climate Change

The distribution of net primary productivity (NPP_A_) values in Qinghai Province reveals higher values in eastern counties compared to western ones, displaying a decreasing trend from southeast to northwest. This pattern is influenced by the regional combination of water and heat conditions shaped by atmospheric circulation and the topography of the Qinghai Plateau [36], transitioning from warm and humid in the southeast to cold and arid in the west. This study affirms that climate factors play a significant role in both the recovery and degradation of vegetation, with climate-driven areas constituting 64.6% of the total area. This underscores climate factors as the primary driving force for vegetation NPP in Qinghai Province. The observed distribution trend of NPP_A_ aligns with potential NPP_P_, decreasing from southeast to northwest, reinforcing that climate factors predominantly impact vegetation NPP_A_ variation. The increasing trend in average NPP_A_ per unit area may be linked to the “warm-wetting” trend proposed in the context of global climate change [37]. This study indicates that 87% of Qinghai Province experiences an increase in vegetation NPP, with 73% attributed to climate factors, supporting this speculation. With the ongoing “warm-humid” trend [37], vegetation NPP_A_ in Qinghai Province is expected to continue increasing, providing an opportunity to leverage climate change for enhanced net primary productivity and alleviating ecosystem carrying pressures.

### 4.3. Potential for Livestock Carrying Capacity Growth

This study reveals that 95% of Qinghai Province has untapped potential for livestock carrying capacity growth, presenting a significant opportunity for enhancing productivity by nearly doubling the current potential of climate resources. To capitalize on this potential, it is recommended to implement degraded grassland restoration projects, particularly in counties with favorable hydrothermal conditions. Counties such as Huzhu, Ledu, Hualong, Minhe, Ping’an, Huangzhong, Xunhua, Gonghe, Guide, Guinan, Tongde, Xinghai, Wulan, Dulan, and Maqin exhibit high potential for livestock carrying capacity growth. These projects aim to optimize pastoral production conditions and maximize climate resource utilization for increased grassland productivity. Embracing modern livestock management practices is crucial for the sustainable development of animal husbandry [28]. This study identifies that the livestock carrying capacity value under climate potential in Qinghai Province in 2018 aligns well with the maximum theoretical livestock carrying capacity estimated by previous researchers [28], serving as a warning threshold for local carrying capacity. Exceeding this threshold could lead to severe impacts on the ecological environment and sustainable development. Despite human activities having a smaller impact on vegetation NPP compared to climatic factors, controlling grazing intensity remains essential to maintain, stabilize, and enhance the regional ecosystem’s carbon sink function.

### 4.4. Methodology

In estimating NPP_P_, only the Zhou Guangsheng model was selected for calculation. Multiple climatic productivity models were not employed to estimate climatic NPP. Instead, the Zhou Guangsheng model was chosen based on its relatively superior performance in prior applications of climatic productivity models within the Three-River Source Region. However, the accuracy of the estimation results still requires improvement and optimization. The comparison between the CASA model simulated values and the measured values of unfenced grassland in this study yielded an R-squared value of 0.3633, which is higher than the R-squared value obtained by Zhang Yonghong [26] using the Zhou Guangsheng model. The relatively low R-squared value in this study may be associated with the limited sample size.

The data products utilized in this study exhibit inherent limitations. The selected remote sensing data products, characterized by relatively coarse spatial resolution, may introduce discrepancies when compared to ground-based vegetation information. For meteorological data, due to the sparse distribution of meteorological stations in western Qinghai Province, the derived results may not fully capture the actual climatic conditions observed in situ. The variation in NPP_P_ results from the combined effects of multiple environmental factors. However, due to the lack of relevant data, this study focused solely on the influence of temperature and precipitation on vegetation productivity, neglecting the impacts of other environmental factors on NPP_P_.

While the method used in this study to evaluate the impact of climate change and human activities on vegetation NPP is simple and easy to calculate, careful consideration of the formula is crucial for accurate results. Further research is warranted to estimate the influence of future climate change on vegetation productivity and address associated uncertainties.

### 4.5. Ecological Management

The eastern, central, and southern regions of Qinghai Province exhibiting increased vegetation NPP align closely with the distribution of alpine meadows and forest ecosystems characterized by superior hydrothermal conditions. This spatial congruence suggests that climate warming and increased precipitation may enhance vegetation recovery by prolonging the growing season or improving photosynthetic efficiency. In contrast, northwestern counties such as Golmud and Mangya, where NPP declines were observed, are situated in the arid Qaidam Basin. The sparse vegetation in these areas may be more sensitive to mining development, where human disturbances can readily exceed ecological thresholds. In climate-dominated vegetation restoration zones (eastern/southern regions), adaptive management should be prioritized, such as dynamically adjusting grassland carrying capacity to align with climate-driven productivity fluctuations. In human activity-dominated degradation zones (northwestern regions), strict limitations on industrial mining activities must be enforced, and ecological restoration projects (e.g., sand fixation) should be initiated.

## 5. Conclusions

This study employs the CASA model to analyze the impact of climate change and human activities on Qinghai province’s vegetation ecosystem from 1982 to 2018. The key findings are as follows:

Climate Dominance: Climate factors are the primary driver of vegetation net primary productivity (NPP), influencing 64.6% of the total area. Human activities contribute to 34.3%, while 2% remains unchanged.

Increased NPP: Climate factors drive a substantial increase in vegetation NPP, covering 87% of the total area. Within this, 73% is influenced by climate factors, spanning all counties in Qinghai. Human activities contribute to the remaining 27%, mainly in the northeastern and southern parts.

Decreased NPP: Human activities are the main factor causing a decrease in vegetation NPP, affecting 11% of the total area. While 1% is driven by climate factors in the northwest of Mangya County, 99% is attributed to human activities in Golmud, Mangya, and Dulan counties in the northwest.

Livestock Capacity Growth: From 1982 to 2018, the livestock carrying capacity in Qinghai province displays an upward trend, with a growth potential per unit area increasing from 0.38 SHU/ha in 1982 to 0.56 SHU/ha in 2018. In 2018, 95% of the total area exhibits positive growth potential, with 9% surpassing 1 SHU/ha, primarily in the eastern part of Qinghai province.

Ecological management and future research directions: In climate-dominated vegetation restoration zones (eastern/southern regions), adaptive management should be prioritized, such as dynamically adjusting grassland carrying capacity to align with climate-driven productivity fluctuations. In human activity-dominated degradation zones (northwestern regions), strict limitations on industrial mining activities must be enforced, and ecological restoration projects (e.g., sand fixation) should be initiated. Future studies should prioritize integrating socioeconomic datasets and employing machine learning models to enhance the predictive capability of analyses.

## Figures and Tables

**Figure 1 biology-14-00494-f001:**
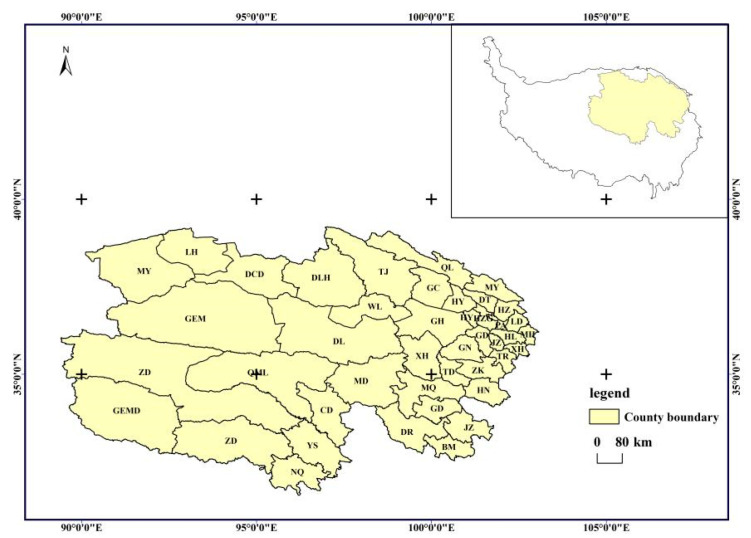
County administrative map of Qinghai Province.

**Figure 2 biology-14-00494-f002:**
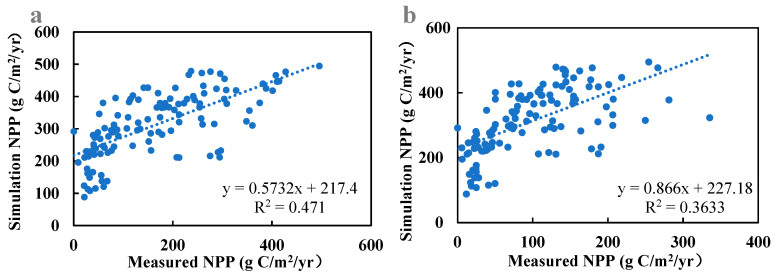
Verification of NPP simulation results. Note: (**a**) represents the comparison between CASA model simulated values and measured values inside the fences, while (**b**) represents the comparison between CASA model simulated values and measured values outside the fences.

**Figure 3 biology-14-00494-f003:**
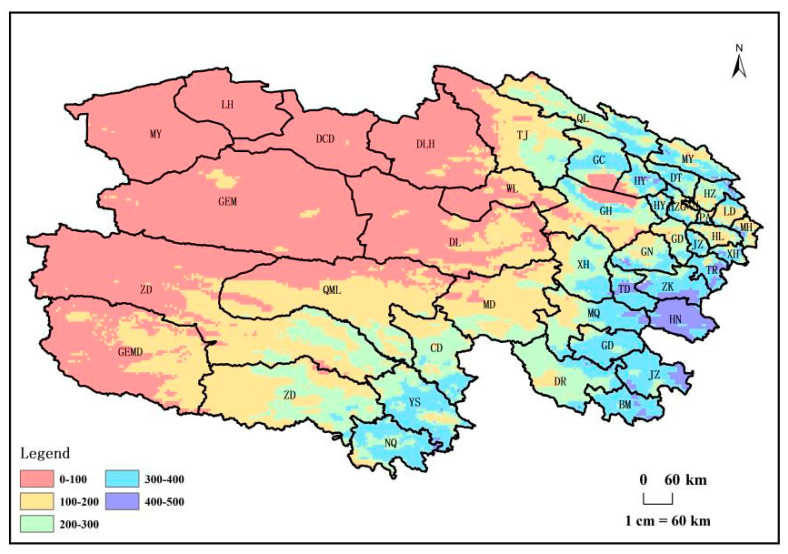
Spatial distribution of the average annual NPP_A_ from 1982 to 2018.

**Figure 4 biology-14-00494-f004:**
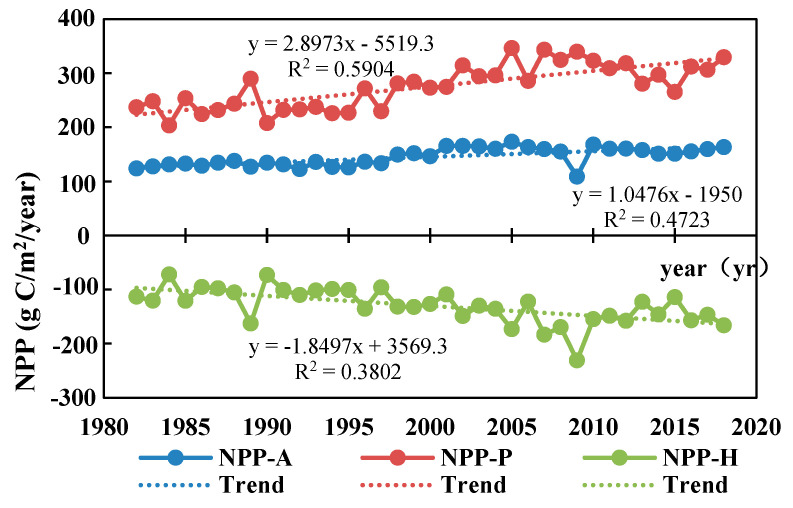
Interannual variations in NPP_A_, NPP_P_, and NPP_H_ from 1982 to 2018.

**Figure 5 biology-14-00494-f005:**
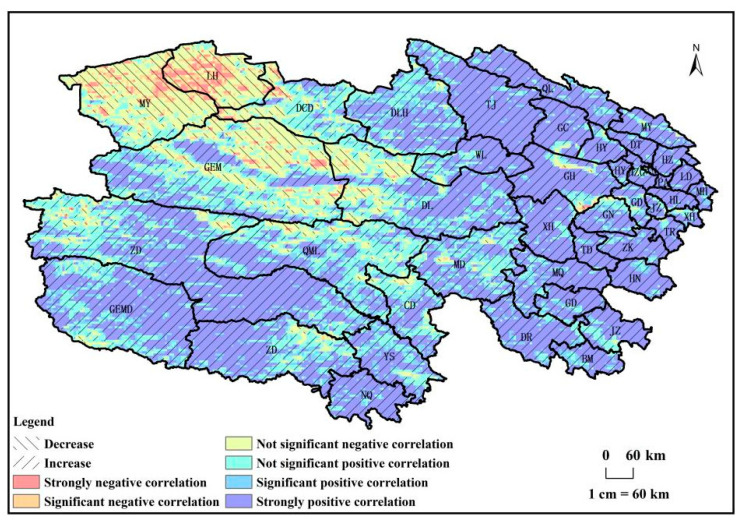
NPP_A_ change trend of vegetation from 1982 to 2018 in Qinghai Province.

**Figure 6 biology-14-00494-f006:**
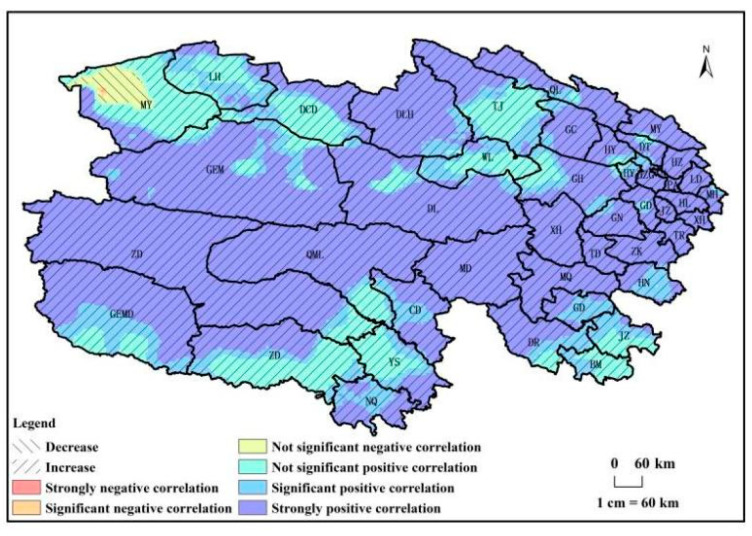
NPP_P_ change trend of vegetation from 1982 to 2018 in Qinghai Province.

**Figure 7 biology-14-00494-f007:**
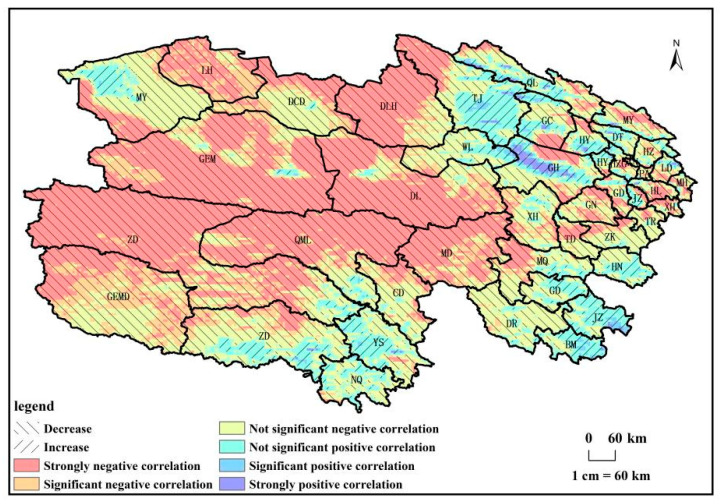
NPP_H_ change trend of vegetation from 1982 to 2018 in Qinghai Province.

**Figure 8 biology-14-00494-f008:**
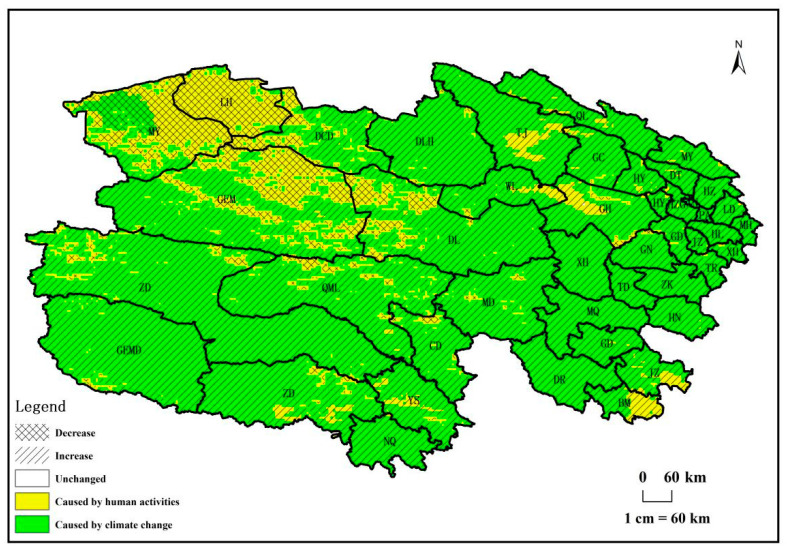
Spatial distributions of different causes of vegetation NPP changes from 1982 to 2018 in Qinghai Province.

**Figure 9 biology-14-00494-f009:**
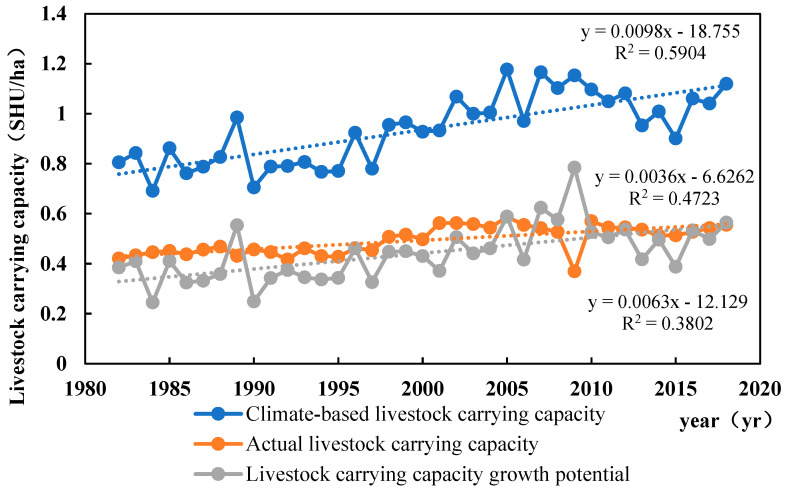
Trends of interannual changes in climate carrying capacity, actual carrying capacity, and growth potential of carrying capacity from 1982 to 2018.

**Figure 10 biology-14-00494-f010:**
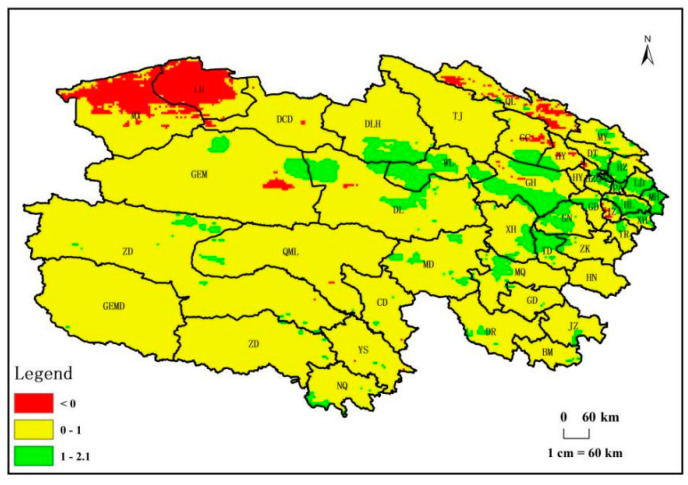
Distribution of growth potential of livestock carrying capacity in Qinghai Province in 2018.

**Table 1 biology-14-00494-t001:** Assessment method of relative effects of climate change and human activities on vegetation NPP.

Change State	Scene	Discrimination Method	Leading Drivers of Change
SA = 0 Vegetation remains unchanged	1		NPPA remains unchanged
SA > 0 Vegetation restoration	2	|SP| > |SH|	Climate factors lead to an increase in NPPA
	3	|SP| < |SH|	Human activities lead to an increase in NPPA
SA < 0 Vegetation degradation	4	|SP| > |SH|	Climate factors lead to a decrease in NPPA
	5	|SP| < |SH|	Human activities lead to a decrease in NPPA

## Data Availability

For this study, remote sensing vegetation index data were sourced from the China Scientific Data Platform (http://data.tpdc.ac.cn/zh-hans/ accessed on 18 May 2022), while meteorological data were supplied by the Qinghai Institute of Meteorological Sciences. Vegetation type data were derived from the GlobeLand30-2010 dataset, downloaded from the National Geospatial Information Resources Catalog Service System. Field measurements originated from monitoring stations operated by Qinghai Province’s meteorological department.

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
