# Peer review of "Qinghai Province (Tibetan Plateau): Quantifying the Influence of Climate Change and Human Activities on Vegetation Net Primary Productivity and Livestock Carrying Capacity Growth Potential"

_biology, 2025, doi:10.3390/biology14050494_

Round 1
Reviewer 1 Report
Comments and Suggestions for Authors
Comments and Suggestions for Authors
The manuscript analyzes the impact of climate and human activities on net primary productivity of vegetation in the Tibetan Plateau and estimates the potential for livestock forage capacity.
General concept comments
Article: The manuscript appears complete and logically structured. I would like to draw attention to one aspect, which is of a recommendatory nature. The authors often mention climatic and anthropogenic factors, but do not disclose the specific causes that influence NPP.
Specific comments
- Line 83. CASA - provide an explanation of the abbreviation and a link to the model in the manuscript.
- 2. Lines 380-385. Dear authors, please clarify what you mean by climatic factors? Of course, in addition to temperature and precipitation. And what specific climatic conditions influenced NPP? What anthropogenic activities were carried out in the study area?
Author Response
Reply to the Reviewer1
Comments 1:
Line 83. CASA - provide an explanation of the abbreviation and a link to the model in the manuscript.
Response 1:
Dear Professor. Thank you for your valuable feedback. We have made revisions in the revised manuscript. “with actual NPP changes computed using the Carnegie-Ames-Stanford Approach (CASA, https://www.nasa.gov/casa-homepage) model in Qinghai Province.”. Line 86-87.
Comments 2:
Lines 380-385. Dear authors, please clarify what you mean by climatic factors? Of course, in addition to temperature and precipitation. And what specific climatic conditions influenced NPP? What anthropogenic activities were carried out in the study area?
Response 2:
Dear Professor. The climate factors mentioned in this paragraph refer to climate change, which is an abstract concept. Of course, it mainly refers to the two influencing factors of temperature and precipitation. We did not use other influencing factors to simulate NPP-P, and did not involve any specific climate factors other than temperature and precipitation. Of course, using only two climate factors, temperature and precipitation, to simulate climate change is a drawback that needs improvement in this article, which has been discussed in the discussion section. Human activities are also an abstract concept in our quantitative research, and we have not studied any specific human activity. The research method is based on references.

Reviewer 2 Report
Comments and Suggestions for Authors
I have the following comments and suggestions for including information and improvements:
Abstract
- The abstract would benefit from a more structured presentation should be enriched the objectives, methodologies, and key quantitative results explicitly to better orient the reader.
- The authors should revise the list of Keywords by adding terms that are not already mentioned in the title.
Introduction
- The literature review should be enriched by integrating more recent literature on the impacts of climate change on vegetation net primary productivity
- The authors are strongly recommended to integrate recent advancements in the literature to provide clearer context for the research question
- The authors should clearly state the existing methodological gaps that this study aims to address
Materials and Methods
- Include a comprehensive flow chart that clearly delineates all steps from data collection through to analysis.
- Provide detailed justification for the selection of CASA model, including its advantages over other models, explicitly referencing there to Qinghai Provinces context.
- Detail parameterization and assumptions of models clearly, including explicit mention of fixed parameters.
- The validation strategy for the CASA model needs clarification.
Results
- Enhance explanations and interpretations of spatial patterns and trends (Figures 3-8), particularly clarifying ecological and management implications.
- Discuss explicitly the variability and potential uncertainties in your model results, particularly those identified in model validation (Figure 2).
Discussion
- The Discussion section would benefit from a clearer structure and more focused articulation of the studies scientific contributions.
- Expanding comparative analysis with a more detailed comparison with previous studies is essential.
- I strongly recommend including a dedicated subsection that compares the study’s methodological approach.
- Clearly discuss potential ecological and land-use management implications derived from your findings, such as grazing management or restoration priorities
- The limitations of the study should be explicitly addressed in dedicating a subsection, and include information about models, addressing data quality, scale, and potential methodological biases.
Conclusions
- Clearly articulate the implications of your findings for ecological management, policy-making, and future research directions.
- Highlight how the methodology and insights from this study could be applicable to similar regions and/or ecological contexts globally.
- I strongly recommend including more specific recommendations for future research, such as integrating socioeconomic datasets, and employing machine learning models.
Figures and Tables
- Improve the resolution and readability of all figures (Figures 3, 5, 6, 7, 8, and 10), including adding coordinate grids and correcting overlaps between axes and graph elements. Particularly figures 2 and 4 should be revised due to their low resolution and the problematic overlap between the Y-axis labels and graphical elements, which hampers data readability.
- Figures 2 and 4 should be revised due to their low resolution and the overlap between the Y-axis labels and graphical elements.
- Enhance figure captions with more descriptive details (Figures 1, 3, 4, 5, 6, 7, 8, 9, 10, 11, and 12).
Author Response
Reply to the Reviewer2
Comments 1:
Abstract
The abstract would benefit from a more structured presentation should be enriched the objectives, methodologies, and key quantitative results explicitly to better orient the reader.
The authors should revise the list of Keywords by adding terms that are not already mentioned in the title.
Response 1:
Dear Professor. Thank you for your valuable feedback. We have made revisions in the revised manuscript. We have adopted a more structured expression in the abstract section. At the same time, we modified the keywords.
Comments 2:
Introduction
The literature review should be enriched by integrating more recent literature on the impacts of climate change on vegetation net primary productivity
The authors are strongly recommended to integrate recent advancements in the literature to provide clearer context for the research question
The authors should clearly state the existing methodological gaps that this study aims to address
Response 2:
Dear Professor. At present, there is relatively little quantitative research on NPP in this research field. The background has been explained in the introduction.This study addresses the gap by juxtaposing the characteristics of climate-induced potential NPP changes, computed using the Zhou Guangsheng model, with actual NPP changes, calculated via the CASA model..
While prior studies employ factors like temperature and precipitation, using correlation analyses to describe NPP spatial differentiation, there remains a gap in quantifying climate change and human activity impacts on ecosystems [16]. Qinghai Province, in particular, lacks sufficient research in this regard.(Line78-81).
Comments 3:
Materials and Methods
Include a comprehensive flow chart that clearly delineates all steps from data collection through to analysis.
Provide detailed justification for the selection of CASA model, including its advantages over other models, explicitly referencing there to Qinghai Provinces context.
Detail parameterization and assumptions of models clearly, including explicit mention of fixed parameters.
The validation strategy for the CASA model needs clarification.
Response 3:
Each specific step of data processing is clearly explained in the methods section.
The CASA model has advantages such as high accuracy and ease of operation compared to other models. Zhang Yonghong[26,27] conducted a qualitative study on NPP in Qinghai Province using the CASA model.(Line167-169)
The principle and parameters of the model have been detailed in the article.
The model-simulated net primary productivity (NPP) of vegetation requires validation using the site observation data provided by the meteorological department in Qinghai Province. Compare the NPP values calculated using the CASA model with the actual site values to verify if they match.(Line271-274)
Comments 4:
Results
Enhance explanations and interpretations of spatial patterns and trends (Figures 3-8), particularly clarifying ecological and management implications.
Discuss explicitly the variability and potential uncertainties in your model results, particularly those identified in model validation (Figure 2).
Response 4:
Explanation and clarification have been provided in the article.This section will be improved in the discussion section.
The eastern, central, and southern regions of Qinghai Province exhibiting increased vegetation NPP align closely with the distribution of alpine meadows and forest ecosystems characterized by superior hydrothermal conditions. This spatial congruence suggests that climate warming and increased precipitation may enhance vegetation recovery by prolonging the growing season or improving photosynthetic efficiency. In contrast, northwestern counties such as Golmud and Mangya, where NPP declines were observed, are situated in the arid Qaidam Basin. The sparse vegetation in these areas may be more sensitive to mining development, where human disturbances can readily exceed ecological thresholds. In climate-dominated vegetation restoration zones (eastern/southern regions), adaptive management should be prioritized, such as dynamically adjusting grassland carrying capacity to align with climate-driven productivity fluctuations. In human activity-dominated degradation zones (northwestern regions), strict limitations on industrial-mining activities must be enforced, and ecological restoration projects (e.g., sand fixation) should be initiated.(Line522-534)
We have made revisions in the revised manuscript. Special discussions have been conducted in the discussion section.(Line 556-563)
Comments 5:
Discussion
The Discussion section would benefit from a clearer structure and more focused articulation of the studies scientific contributions.
Expanding comparative analysis with a more detailed comparison with previous studies is essential.
I strongly recommend including a dedicated subsection that compares the study’s methodological approach.
Clearly discuss potential ecological and land-use management implications derived from your findings, such as grazing management or restoration priorities
The limitations of the study should be explicitly addressed in dedicating a subsection, and include information about models, addressing data quality, scale, and potential methodological biases.
Response 5:
The discussion section has been revised. We have specifically discussed methods in the discussion section.(L495-535) The discussion section has been compared with previous research.
Comments 6:
Conclusions
Clearly articulate the implications of your findings for ecological management, policy-making, and future research directions.
Highlight how the methodology and insights from this study could be applicable to similar regions and/or ecological contexts globally.
I strongly recommend including more specific recommendations for future research, such as integrating socioeconomic datasets, and employing machine learning models.
Response 6:
The modification has been completed in the conclusion section.
Comments 7:
Figures and Tables
Improve the resolution and readability of all figures (Figures 3, 5, 6, 7, 8, and 10), including adding coordinate grids and correcting overlaps between axes and graph elements. Particularly
figures 2 and 4 should be revised due to their low resolution and the problematic overlap between the Y-axis labels and graphical elements, which hampers data readability.
Figures 2 and 4 should be revised due to their low resolution and the overlap between the Y-axis labels and graphical elements.
Enhance figure captions with more descriptive details (Figures 1, 3, 4, 5, 6, 7, 8, 9, 10, 11, and 12).
Response 7:
Completed the modifications in the revised manuscript. The overall resolution of the image has been enhanced and the overall image quality has been improved.
Reviewer 3 Report
Comments and Suggestions for Authors
This study assesses the relationships between temporal patterns of net primary productivity (NPP) and climate-vegetation change in a mountainous region. Estimates of NPP are validated against field observations, providing a robust approach to this estimation. I have a few comments to improve this manuscript:
-The number of plots is excessive. I would therefore suggest reducing them by merging them into panels or sending some of them to the appendix.
L67 and L81. Refer to Net Primary Productivity using its abbreviation
L83-88 Please list these objectives and include hypotheses for each of them (if possible).
L291 A R-squared value of 0.36 is relatively low, please discuss this result
Figure 4. In the legend, please explain the pointed lines represent the trend
L330-333. This information is methodological, please move to the Methods section.
Author Response
Reply to the Reviewer3
Comments 1:
-The number of plots is excessive. I would therefore suggest reducing them by merging them into panels or sending some of them to the appendix.
Response 1:
Dear Professor. Thank you for your kind feedback.The layout is arranged according to the editor's instructions.
Comments 2:
L67 and L81. Refer to Net Primary Productivity using its abbreviation
Response 2:
Dear Professor. Thank you for your valuable feedback. We have made revisions in the revised manuscript.
Comments 3:
L83-88 Please list these objectives and include hypotheses for each of them (if possible).
Response 3:
It's a bit challenging. The NPP simulation values of the model are reflected in the results section.
Comments 4:
L291 A R-squared value of 0.36 is relatively low, please discuss this result
Response 4:
We have made revisions in the revised manuscript. Special discussions have been conducted in the discussion section.(L482-502)
Comments 5:
Figure 4. In the legend, please explain the pointed lines represent the trend
Response 5:
The modification has been completed in Figure 4.
Comments 6:
L330-333. This information is methodological, please move to the Methods section.
Response 6:
This section has been moved to the Methods section.